# Electromagnetic Assessment of UHF-RFID Devices in Healthcare Environment

Victoria Ramos [1,*], Oscar J. Suárez [2], Samuel Suárez [3], Víctor M. Febles [3], Erik Aguirre [4], Patryk Zradziński [5], Luis E. Rabassa [3], Mikel Celaya-Echarri [6], Pablo Marina [1], Jolanta Karpowicz [5], Francisco Falcone [4,6] and José A. Hernández [3]

1   Instituto de Salud Carlos III, 28029 Madrid, Spain
2   Dirección General de Telecomunicaciones y Ordenación de los Servicios de Comunicación Audiovisual, 28071 Madrid, Spain
3   Engineering and Telematic Department, Hospital Universitario de Canarias, 38320 La Laguna, Spain
4   Electrical, Electronics and Communications Department, Universidad Pública de Navarra, 31006 Pamplona, Spain
5   Central Institute for Labour Protection–National Research Institute (CIOP-PIB), 00-701 Warszawa, Poland
6   School of Engineering and Science, Tecnologico de Monterrey, Monterrey 64849, NL, Mexico
*   Correspondence: vramos@isciii.es; Tel.: +34-918-222-128

**Abstract:** In this work, the evaluation of electromagnetic effect of Ultra High Frequency Radio Frequency Identification (UHF-RFID) passive tags used in the healthcare environment is presented. In order to evaluate exposure levels caused by EM field (865–868 MHz) of UHF-RFID readers, EM measurements in an anechoic chamber and in a real medical environment (Hospital Universitario de Canarias), as well as simulations by 3D Ray Launching algorithm, and of biophysical exposure effects in human models are presented. The results obtained show that the EM exposure is localized, in close vicinity of RFID reader and inversely proportional to its reading range. The EM exposure levels detected are sufficient to cause EM immunity effects in electronic devices (malfunctions in medical equipment or implants). Moreover, more than negligible direct effects in humans (exceeding relevant SAR values) were found only next to the reader, up to approximately 30% of the reading range. As a consequence, the EM risk could be firstly evaluated based on RFID parameters, but should include an in situ exposure assessment. It requires attention and additional studies, as increased applications of monitoring systems are observed in the healthcare sector—specifically when any system is located close to the workplace that is permanently occupied.

**Keywords:** UHF-RFID; occupational exposure; public health; SAR; 3D-RL; electromagnetic risk; hospitals; 2D contour maps

## 1. Introduction

The application of the Information and Communication Technologies (ICT) that guarantee smart and seamless assistance has been significantly increasing and, thus, the health sector has become dependable on these technologies. To quote a relevant example, a reference should be made to the tracking system that is based upon short-range radio frequency, Ultra High Frequency—Radio Frequency Identification (UHF-RFID). This system is used to aid the identification process of materials, instrumentation and people at healthcare centers.

At the outset, the monitored objects are assigned a digital identity, with the use of labels attached to the objects that contain written individual number, as replaced or supplemented by barcodes, which are optically read by scanners (using infrared laser beams). With regard to the use of optical technologies, long-term and substantial human efforts are required for an object to be subject to individual processing, as an object needs to be approached

to the scanner each time and a bar code needs to be located precisely within the optical scanner line.

The RFID is a widely used wireless technology for Automatic Identification and Data Capture (AIDC); it has been commonly applied to identify and track tags attached to specific objects. The RFID readers obtain and write data on tags, which are most frequently "passive tags", i.e., small electronic circuits which are supplied (charged) by the production of electromagnetic energy of the electromagnetic field (EMF) transmitted by the reader of RFID system.

Smart solutions for medical purposes (e-Health) are globally implemented as Smart Health [1] solutions. In this context, RFID technology enables objects to retain to their identity, similar to a unique serial number. The wireless RFID data exchanged are processed to provide real-time data in the monitoring systems. Usually, fixed readers and handheld readers are employed and, to this extent, from the point of view of the environmental electromagnetic impact, the following are most often recognized to be fit for their intended purpose: Manually Operated (MO) readers or Autonomous (A) readers. A schematic of performance of performance of an UHF-RFID system used in healthcare centers [2] is depicted in Figure 1.

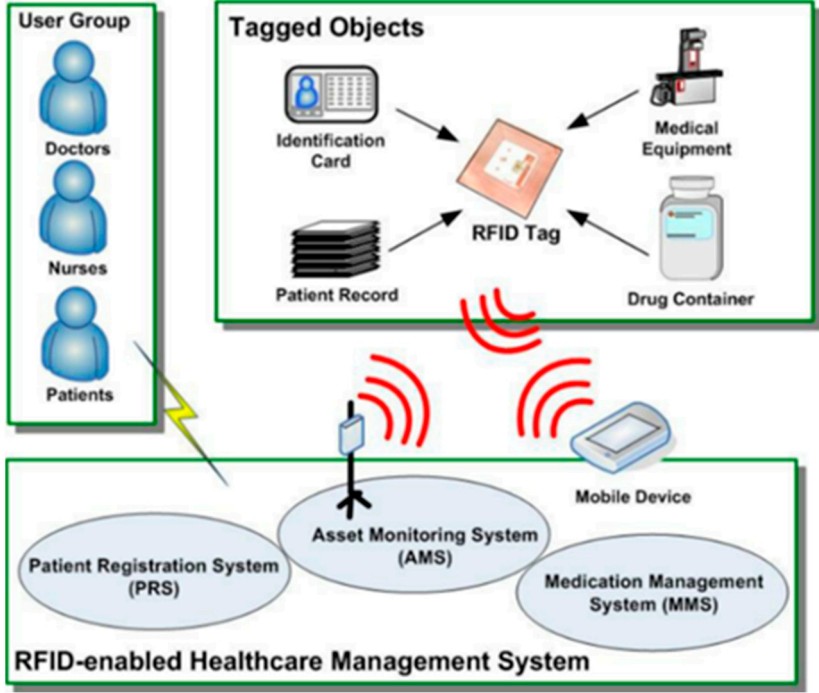

**Figure 1.** RFID performance scheme at a health center [2].

What we may observe today is that the environment and individuals are more and more exposed to new electromagnetic conditions, with no relevant provision made therefore in the legislation on human exposure or sufficient previous studies on their longstanding effect, considered low intensity exposure effects or operation of medical devices in indoor environments [3–9]. The electromagnetic assessment of the operation of the UHF-RFID system, that operates at the 865–868 MHz range, being in fact similar to the frequency of the EMF emitted by mobile communication GSM systems, and is essential for underpinning all aspects in response to a high demand for dynamic wireless applications in the healthcare environment.

For these technologies to be developed, manufactured and deployed further on, a need for signals, devices, antennas and complete systems to be measured and analyzed with respect to RF exposure nearby has emerged. The Electromagnetic Interference (EMI) may cause severe problems to electronic devices, and can imply dangerous consequences,

if medical devices are affected. In relation with the effect of exposure on humans, the studies regarding the risk posed by mobile communication devices to health prove of relevance here [8,10,11]. In all cases, the electromagnetic environmental impact and the related risk can be mitigated when the relevant preventive measures are undertaken, as shown in [12,13].

The exposure levels recorded for the UHF-RFID system are subject to examination with the aim of verifying compliance with relevant laws and regulations; for example, the European general public recommendation 1999/519/EC, the European Directive 2013/35/EU on workers EMF exposure, EMC, RED or MED directives, using relevant European Standards or legislation transposing international legislation into legal system in particular countries, that apply to the electromagnetic environmental impact on safety of patients, attending visitors, workers or various electronic devices that have been used in the healthcare environment. In brief, in compliance with the laws and regulations concerning the minimum safety requirements as set forth in by international legislation, it has been assumed that humans should be protected against thermal effects caused in the body by exposure to the RF EMF. Considering the fact that sensitivity to these exposure effects differs in population, vulnerable populations should be protected against such exposure more restrictively (for example, pregnant women or users of medical implants).

The laws and regulations on the limitation of the EMF exposure levels are usually divided into those that refer to the so-called limits regarding workers' EMF exposure (which may be applied when the exposed humans are notified of and trained on the results of periodical assessment of exposure, electromagnetic hazards expected at the workplace, as well as any necessary preventive measures that was applied there; additionally, the volume and location of the space affected by EMF at the workplace should be accurately designated, protected against an access by unauthorized individuals, as well as all the actions undertaken by the employer are sufficiently documented in an appropriate manner). Exposure limits applicable to general public exposure are lower than the workers' exposure, as they are applicable to the population that may not be informed of electromagnetic hazards, and include vulnerable individuals; and, no preventive measures are required to be applied to reduce the expected electromagnetic hazards. However, the description of both kinds of exposure limits includes also the information on the exceptions to the rule—for the exposure lower than the specified limits; electromagnetic hazards are not excluded where the EMF affects vulnerable persons or electronic devices.

In general, the electronic devices may be almost completely immune to electromagnetic disturbance (even if its effect is as strong as an intentional military electromagnetic attack); but to achieve such immunity, the cost-generating design of the device is required (generating even very expensive production of highly immune devices). Various lower-cost electronic devices, that are at the same time much less immune to electromagnetic disturbances, are used in everyday life and work, depending upon their function and intended use. At the minimum immunity of low-cost electronic devices, they can actually operate in an appropriate manner when affected by EMF of the typical characteristic expected in the environment of their intended use; and, for the case of RF EMF impact on the devices used today, it is the level of exposure at the level of small fraction of general public exposure limits provided by the aforementioned European legislations. Taking into consideration the relevant role that the medical devices play today, including implanted devices such as cardiac stimulators, by applicable technical regulations (like, for example, technical regulations, namely EMC, RED, LVD or MED directives) it is required that they have better electromagnetic immunity than the basic immunity required for various low-cost electronic devices for everyday use. A device manufacturer delivers a relevant declaration on the level of electromagnetic immunity, along with the operating manual, e.g., in a section with the description of environmental conditions, safety, or EMC.

A relevant study of the analysis of the electric field (E-field) parameters is made, for example, as regards the exposure to spatial distribution of the results of the 3D Ray Launching (3D-RL) technique simulation, as displayed via 2D contour maps, as being

processed with the use of a relevant MATLAB® algorithm [13,14]. Some other studies of the E-field parameters were based upon the scenario which entailed the far-field conditions, and applicable at least a several wavelength away from the emitting antenna [15,16]. There are also studies carried out whereby there were performed the measurements or numerical simulations of the Specific Energy Absorption Rate (SAR), and whereby the information on the biophysical effects of the EMF exposure in the human body was obtained, as well as the results of the research assessment with regard to the relevant limits related to the protection against thermal effects caused by EMF influence, that have been accurately determined under applicable laws and regulations and exposure assessment guidelines described in the analysis of the research papers, for example [17–22].

## 2. Materials and Methods

The purpose of the RF EMF measurements was to assure the compatibility between wireless systems installed at a healthcare center [23,24]. In order to analyze their use in the healthcare environment and the influence of the UHF-RFID system at the 865–868 MHz frequency range, several measurement procedures have been employed. In situ measurements are performed, among other things, to:

- determine RF exposure levels from the Equipment Under Test (EUT) and ambient sources,
- consider applicable regulations (exposure limits), as observed or extrapolated to maximum or actual maximum levels; or
- to monitor the RF exposure levels, even if they are well below the exposure limits that apply to ensure the protection against thermal effects of EMF influence.

In order to accomplish the objectives set for in situ measurements, it is required that frequency selective assessment be carried out whereby the contributions of relevant sources of interest to the total RF exposure will be determined, including other exposures from radio-communication systems such as mobile phones or internet access.

The projects were subject to the two assessment procedures, i.e., the first at the Hospital Universitario de Canarias (HUC), with the measurements having been performed where the RFID devices had been installed for assessment purpose. Additional measurements were performed in Madrid, where the assessment was made in an anechoic chamber, with the aim being to examine the worst-case scenario. An ad-hoc numerical simulation study was performed by means of: (1) Specific Absorption Rate (SAR) values from exposure near MO-RFID readers, (2) distribution of exposure from AG-RFID reader by means of an in-house 3D-RL algorithm, and (3) 2D contour maps representation [25,26].

### 2.1. UHF-RFID Devices Assessed

At the beginning of the process, there is an in situ analysis performed to identify the source of RF emission within the surrounding area. The in situ RF exposure assessment was performed at the measurement area, with the use of the selected exposure metric, measurement type and measurement techniques to satisfy the measurement objectives. Figure 2 depicts specific devices that were included into the study, i.e., manually operated (MO) and autonomous gate (AG) and autonomous shelf (AS) RFID readers (as specified in Table 1, including passive tags compatible with all the specified readers).

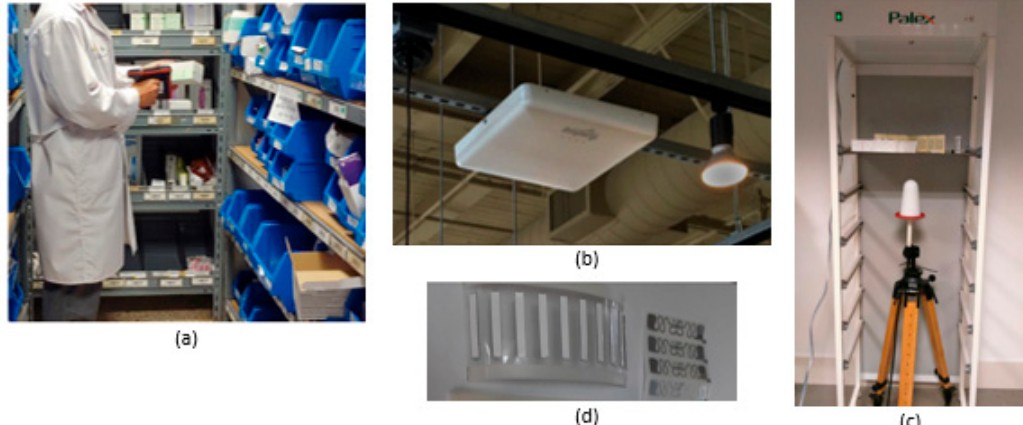

**Figure 2.** The RFID UHF readers used for the purpose of the study and commonly used at hospitals: (**a**) handheld reader AB700 (reader type: MO-RFID); (**b**) fixed xArray Gateway R680 (reader type: AG-RFID); (**c**) shelving DYANE Smartshelf (reader type: AS-RFID), (**d**) passive RFID tags compliant with (**a–c**) readers (type: EPCglobal UHF RFID Class 1 Gen 2).

**Table 1.** The Specifications of RFID UHF readers commonly used at hospitals and used during the study.

| Description | Key Parameters of the RFID UHF Readers Used for the Study | | |
| --- | --- | --- | --- |
| | Handheld Reader AB700 | Fixed xArray Gateway R680 | Shelving DYANE Smartshelf |
| Reader type | Manually Operated (MO-RFID reader) | Autonomous Gate (AG-RFID reader) | Autonomous Shelf (AS-RFID reader) |
| Frequency | 860–960 MHz | 860–960 MHz | 860–960 MHz |
| Air Interface Protocol (compliant tags) | EPCglobal UHF RFID Class 1 Gen 2 [1] ISO/IEC 18000-63 | EPCglobal UHF RFID Class 1 Gen 2 ISO/IEC 18000-63 | EPCglobal UHF RFID Class 1 Gen2 ISO/IEC 18000-63 |
| Max Receive Sensitivity | not available | −82 dBm [0.5 mV/m] | −84 dBm [0.4 mV/m] |
| Tag read sensitivity | from (−22 dBm) to (−10 dBm) [equivalent to the range: 0.5–2 V/m] | | |
| Reading Range | 0–7 m (according to tag and environment) | up to 1.5 m (according to tag and environment) | the entire interior of the cabinet |
| Writing Range | 0–3 m (according to tag and environment) | not applicable | not applicable |
| Transmitted Power | up to 1 W ERP | up to 2 W ERP (EU1) [27] 4 W ERP (EU2) [27] 4 W ERP (FCC) [28] | up to 1 W ERP (EU1) |
| Antenna | 1 antenna | 1 phase array antenna (52 dual-polarized beams in 9 sectors) | 4 expandable to 32 antennas with Speedway Antenna Hub optimized for imping reader antennas (RP TNC connector) |

[1] duty cycle 48–82.3% for interrogator (reader) to tag communications [29].

The handheld readers (AB700 (ETSI)) are gun-shaped transceivers, normally used close to the human body. The RFID is used as shown in Figure 2a and can affect specific parts of the operator's body. It is recommended to make RFID readings with the arm extended to limit the exposure, but it increases the musculoskeletal load experienced by operator [7,8,30]. The relevant technical specifications are specified in Table 1.

The UHF-RFID fixed xArray (Gateway R680 (ETSI)) used for the study is located within at the Hospital Universitario de Canarias (HUC). The platform was installed for the assessment purpose, and the compliance with the requirements set was conducted. The xArray provides real-time identification and localization of the tagged items. It was positioned in the center of a room ceiling, with a room height of approximately 3.5 m, and eventually the housed estimated area was of 8 × 8 m. The Impinj's Item Sense management software triggers and automatically handles the management and monitoring infrastructure. The UHF-RFID device used for the study is presented in Figure 2b and the relevant technical specifications are presented in Table 1.

The device DYANE Smartshelf subject to assessment hereunder is an RFID smart shelving system, that is used for automation of tagged reagent inventory control in operating theatres and clinical laboratories in real time. It is presented in Figure 2c. The process is handled, with the use of the DYANE software, which provides access to parameter reports. The relevant technical specifications are shown in Table 1.

### 2.2. EMF Spatial Distribution and Level Measurements

The comprehensive exposure assessment, source identification, and configuration at a maximum transmission power or actual maximum power was accurately performed for the purpose of this paper. The characteristic features of the RF EMF exposure were studied with respect to its spatial distribution to be denoted in E-field strength (E, in V/m), based on results of measurements or calculations.

### 2.2.1. Hospital Universitario de Canarias (HUC)—Tenerife (Assessment of MO-RFID and AS-RFID Readers)

The measurements were performed, with the use of frequency selective equipment, i.e., a R&S FSH6 spectrum analyzer (measurement uncertainty <1.5 dB, typical 0.5 dB) and an antenna ETS-LINDGREN, Mod.3181 (500–9000 MHz), that are specified in Figure 3.

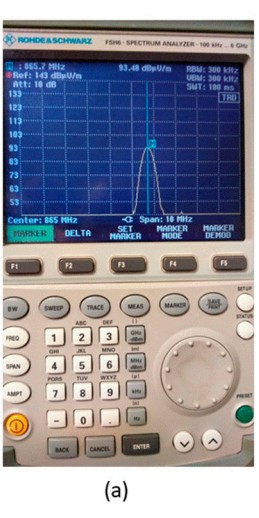

(a)

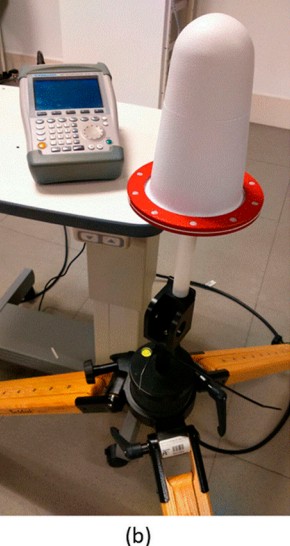

(b)

**Figure 3.** Measurement set up used for in situ measurements of EMF emitted by studied RFID readers: (**a**) Rhode & Schwartz FSH6 spectrum analyzer, (**b**) Antenna ETS-LINDGREN, Mod.3181 on tripod.

The measurements were made at an RFID core frequency (866 MHz), in a "demo" mode of MO-RFID reader operation, whereby undisturbed emission was guaranteed, without a need for human involvement in the procedure.

The measurements were carried out in a wide-open area, a tower located on the 4th floor of the hospital. Measurements were recorded at distances of from 0.25 and 4 m, and at angles between 0° and 180°, rotating in 45° angle steps.

The E-field was assessed near the reader with an antenna located vertically (compliant with a regular position when the MO-RFID reader is used at a "gun-like" position). The exposure scenarios (ESs) include:

- Exposure caused by activation of MO-RFID reader placed on a table, at a 1 m height, with reading tags located at a 1 m height from the floor
- Five angles (degrees): 0, 45, 90, 135, and 180
- Five distances (m): 0.25, 0.5, 1, 2, 4.

An AS-RFID reader was placed inside a room at an approximate height of 3.5 m, without windows and with a door near the wall. Measurements were carried out at 866 MHz and at 1.0 m, 1.3 m and 1.7 m heights from the floor, representing the zone of genitalia, breast and abdomen, and head respectively to the floor, as they are considered sensitive parts of a human body. The levels were also assessed inside the Smartshelf 0.075 m to the rear part, which was a minimum distance of proximity. The fact that there were no doors in the wardrobe can be used to examine the worst-case scenario. When it is located at the patient examination area, they would be exposed rather sporadically and of short duration.

2.2.2. Anechoic Chamber—Madrid Laboratory (Assessment of MO-RFID and AG-RFID Readers)

The measurements of the ERP that involved the worst-case scenario were made at The Dirección General de Telecomunicaciones y Ordenación de los Servicios de Comunicación Audiovisual (Madrid). Table 2 presents the list of the tools that were mainly used for this purpose.

**Table 2.** Laboratory devices used for anechoic chamber UHF-RFID assessment.

| Equipment | Marque-Brand | Model |
|---|---|---|
| EMI Receptor | Rohde & Schwartz | ESIB26 |
| RF Generator | Rohde & Schwartz | SMT02 |
| Log-Period Antenna | EMCO | 3146 |
| Semi anechoic chamber | IRSA | 3 m |
| Software | Rohde &Schwartz | EMC32-E |
| Mast | EMCO | 1050 |
| Table | EMCO | 1060-1.2 |

Measurement uncertainty: ±3.81 dB of E-field strength; at duty cycle: 100%.

The measurements of EMF emission of RFID readers (working band: 865–868 MHz) were carried out in an anechoic chamber with an absorbent on the floor, with the device situated vertically on a support (in case of a MO-RFID reader, also in the presence of an individual to achieve continuous transmission). The E-field strength (E) emitted at distances of 0.5 m, 1.0 m and 3.0 m from the equipment were recorded. The measurement setup is presented in Figure 4. The study includes a concise and precise description of the procedures carried out during the experiment. The complete and detailed information is available in [13].

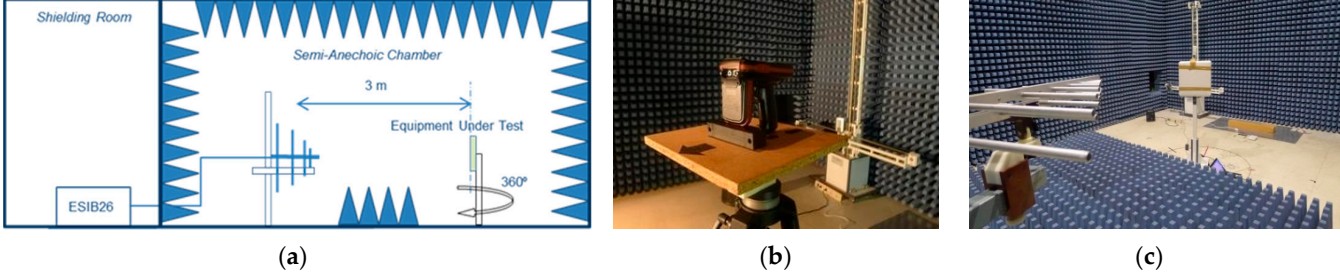

**Figure 4.** Laboratory measurements of the EMF emitted by RFID readers inside the anechoic chamber: (**a**) measurement scenario, (**b**) measurement of a MO-RFID reader, (**c**) measurements of AG-RFID reader.

*2.3. Simulations of Exposure from the AG-RFID Reader by Means of an In-House 3D-RL Algorithm*

An in-house simulation MATLAB® algorithm was used and E-field estimations were obtained. Far-field conditions were supposed during all the processed simulations, and the parameters like dielectric permittivity and conductivity were obtained for specific elements. The simulation results exhibit uncertainty in the 2 dB to 5 dB range, owing to the 3D scenario mapping process.

Specific materials were considered with variables in Table 3. Other simulation parameters are shown in Table 4, apart from free space losses, reflection, refraction or diffraction. The emission was produced by the devices at the same positions as in the real-life scenario. A concise and precise description of the experimental work is presented here. The complete and detailed information is available in [13].

**Table 3.** Material Properties for 3D Ray Launching Simulations [13].

| Material | Conductivity ($\sigma$) [S/m] | Relative Permittivity ($\varepsilon r$) |
|---|---|---|
| Aluminum | 37.8·106 | 4.5 |
| Steel | 7.69·106 | 4.5 |
| Nylon | 0.24 | 1.2 |
| Wood | 0.21 | 2.88 |
| PVC | 0.12 | 4 |
| Polypropylene | 0.11 | 3 |
| Glass | 0.11 | 6.06 |
| Concrete | 0.02 | 25 |
| Rubber | $1 \times 10^{-14}$ | 2.61 |

**Table 4.** Variable Configuration for 3D Ray Launching [13].

| Parameter | Value |
|---|---|
| Operation Frequency | 865–868 MHz |
| Transmitted Power | 0.00316–2 W |
| Horizontal angular resolution ($\Delta\Phi$) | 1° |
| Vertical angular resolution ($\Delta\theta$) | 1° |
| Permitted maximum reflections | 6 |
| Cuboids size (Mesh resolution) | 0.5 m × 0.5 m × 0.5 m |
| Diffraction phenomenon | Activated |

Simulation uncertainty: 2 dB to 5 dB, owing to the 3D scenario mapping process.

The graphic representation of the information on the EM levels is considered a well-known tool for clear depiction of the information as presented in previous works drafted by [6,13,14].

This paper has been prepared with the use of the methodology for the production of 2D contour maps that offer graphic, immediate and accurate representation of the EMF distribution. The data collected were imported using graphic software (Surfer 8) to draw 2D contour maps showing the previously measured field strength values. The selected measurement points are displayed in Figure 5.

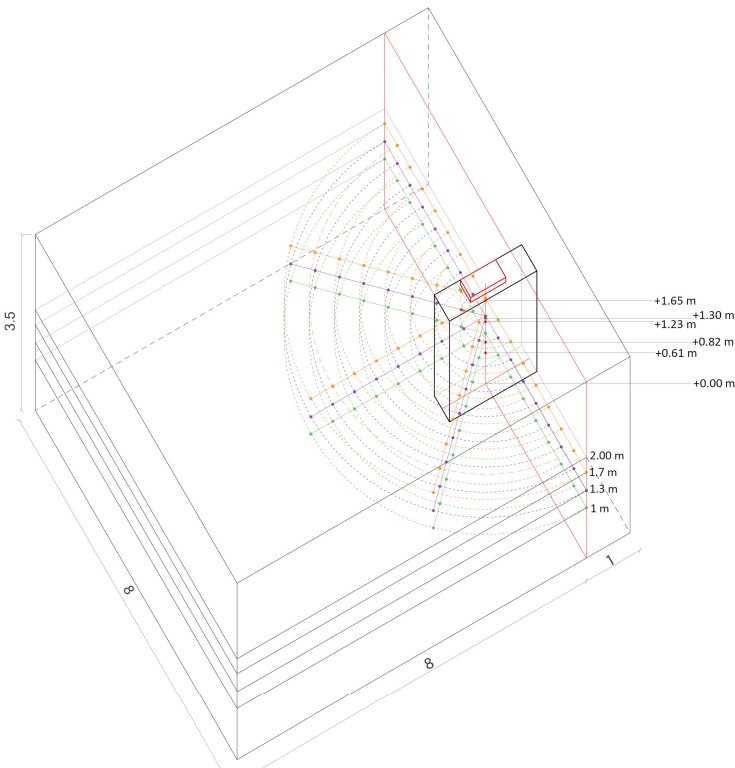

**Figure 5.** Grid and measurement points in a care room of the HUC for the AS-RFID reader assessment.

*2.4. Specific Absorption Rate (SAR) Values from Exposure near MO-RFID Readers*

When individuals approach the source of EMF where near-field exposure conditions are recognized (i.e., in the distance from the source shorter than the wavelength of emitted EMF or even several wavelengths), their exposure has to be assessed using the parameters characterizing the effects of EMF influence in tissues. In case of radiofrequency EMF exposure, the relevant parameter is the metrics of thermal effects—time derivative of the incremental electromagnetic energy (dW) absorbed by an incremental mass (dm), recognized as SAR values expressed in watts of absorbed EMF energy per kilogram of exposed tissue, (W/kg). The SAR is assessed as averaged over the entire exposed body and also as localized SAR averaged in particular smaller parts of the exposed body (over 10 g mass of any continuous tissue—SAR10g) [12]. SAR values are assessed by the numerical modeling of the EMF interaction between the model of EMF source and heterogeneous anatomical body model, usually with the use of Finite-Difference Time-Domain (FDTD) numerical code compliant with the requirements set forth in the relevant international standard [31]. In the international guidelines and applicable laws and regulations relevant limits regarding mentioned SAR, values have been provided aimed for assessment of EMF exposure of workers and general public. As provided for in the relevant international laws and regulations, the SAR must be assessed for all the cases of the localized exposure within reactive near-field [24]. Moreover, under the Directive 2014/53/EU (RED), it is required to take all intended and reasonably foreseeable operating conditions of radio equipment use into account for the exposure assessment [32].

The parameters of a typical MO-RFID reader used for our simulation are presented in Table 1. As regards the exposure scenarios applied for the purpose of the study, the antenna was located vertically (similarly to the position of an external antenna in a MO-RFID reader shown in Figure 2). The exposure scenarios include, as presented in Figure 6:

-    Exposure of an operator of a MO-RFID reader grasping a gun in the hand,
-    Exposure of people approaching an operator of an MO-RFID reader—scanned person or bystander.

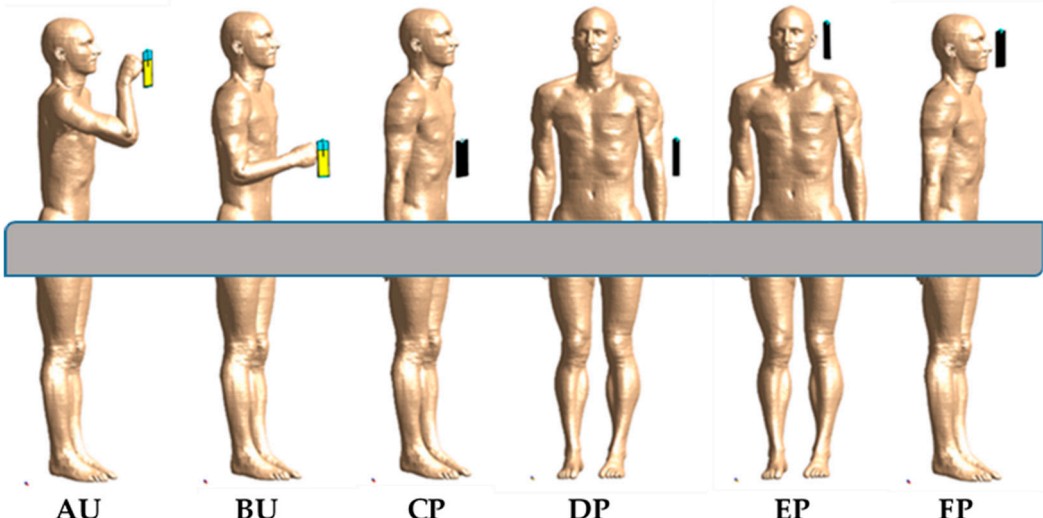

**Figure 6.** The exposure scenarios: (1) exposure of an operator grasping a MO-RFID reader (AU—in front of the face; BU—in front of the chest, (2) exposure of an individual approaching a RFID gun used as in AU or BU scenario (CP—in front of the chest; DP—at the side of the body, at the height of the chest; EP—at the side of the body, at the height of the head; FP—in front of the face). The anatomical numerical model of operator—Duke developed by the ITIS Foundation (over 300 tissues/organs: muscles, fat, bones, nervous tissues, heart, liver, skin, etc.).

Based on the principles of the electromagnetic hazard management system, the EMF exposure of a reader operator is assessed with respect to the exposure limits set for the worker exposure (when an operator is an informed and qualified worker and the relevant prevention measures with respect to electromagnetic hazards are applied and accurately evidenced), or with respect to the general public exposure limits (when a gun is used by a person assuming to be without a regular employment contract, i.e., with no understanding of electromagnetic hazards and where prevention measures are not applied).

Similarly, the exposure of a scanned person or a bystander is assessed in terms of the exposure limits set for the general public exposure or worker exposure (depending upon the aim of their presence nearby a activated reader and applied electromagnetic hazards management system).

Numerical simulations were carried out with the use of anatomical numerical male model and Sim4Life software (Zurich Med Tech, Switzerland) and Finite Difference Time-Domain solvers (FDTD). The FDTD method is a solution of Maxwell's curl equation in the time domain [31]. The uncertainty of the numerical simulations (i.a. related to the model of EMF source, dielectric properties of the model of human body and model resolution (staircase approximation)) was estimated as not exceeding $\pm 1$ dB (K = 1), within the range compliant with the state of the art in the field [33,34].

An accurate description of the work is presented here, however additional information is available in [12].

## 3. Results

This is here to focus on the electromagnetic impact of the UHF-RFID system upon the healthcare environment (studied through measuring and/or modeling the EMF parameters or modeling the effects of EMF exposure), to include:

(a)　Measurements of E-field values at a platform at the HUC (evaluation of EMF emitted by MO-RFID and AS-RFID readers);

(b)　Measurements of the E-field values at an anechoic chamber, that have been taken as for a worst case (evaluation of EMF emitted by MO-RFID and AG-RFID readers);

(c)　Simulations of the SAR values, using the FDTD method (evaluation of effects of exposure to EMF emitted by MO-RFID reader);

(d)   Simulations of the E-field values, using an in-house 3D-RL algorithm and 2D contour maps representation (evaluation of the EMF emitted by the AG-RFID reader).

*3.1. Hospital Universitario de Canarias (HUC)—Tenerife (Measurement-Based Assessment)*

An extensive catalogue of opportunities for the RFID technology to serve as an intended use also involves a variety of the EMF exposure options on the side of device operators, along with other persons that may be present at the site.

### 3.1.1. EMF Exposure near MO-RFID Reader

Radiation pattern, (to include maximum values from measurements for horizontal and vertical positions of measuring antenna and adjusted RFID antenna position) was sampled out around antenna starting in front of UHF RFID antenna with angle step of 45º, to cover major, lateral, and back radiation.

The E-field distribution was evaluated along three axes passing through the center of the antenna of the MO-RFID reader (gun): in the front and back of the reader plane—perpendicular to it and at the side of the reader—perpendicular to the reader's edge at some distances of the reader center. The results were normalized to an output power equal to the case considered in the numerical simulations (1 W). Measurements obtained in Torre Norte 4 Floor may be observed in Table 5.

**Table 5.** The E-field measurements results obtained near the MO-RFID reader: frequency—866 MHz, location—HUC, Torre Norte 4 Floor.

| Distance (m) | E-Field Peak Value [V/m] | | | | |
|---|---|---|---|---|---|
| | Angle (º) | | | | |
| | 0 | 45 | 90 | 135 | 180 |
| 0.25 | 0.117 | 0.117 | 0.117 | 0.118 | 0.118 |
| 0.50 | 0.118 | 0.118 | 0.118 | 0.118 | 0.119 |
| 1.00 | 0.119 | 0.119 | 0.112 | 0.109 | 0.105 |
| 2.00 | 0.099 | 0.070 | 0.066 | 0.090 | 0.093 |
| 4.00 | 0.062 | 0.058 | 0.050 | 0.040 | 0.042 |

Level measurement uncertainty <1.5 dB, typical 0.5 dB.

Most of E-field intensity levels were in the range from 0.040 V/m to 0.090 V/m. Nevertheless, a peak value of 0.210 V/m was registered at the angle 0°, at the height of 1.0 m and at a distance of 1.5 m. Lower values of 0.137 V/m and 0.116 V/m were registered at the 180° angle, at 1.5 and 2.0 m, respectively. The peak value of 0.210 V/m corresponds to front position: MO-RFID reader and antenna are at the height of 1 m and between them, there are 1.5 m and the lowest at the 180° angle.

In addition, a peak of 0.190 V/m was registered at the height of 1.7 m, at the 0° angle and at a distance of 1 m, obtaining lower values, 0.155 V/m and 0.127 V/m, at the 180° angle, at 1.0 and 1.5 m, respectively. As for the previous measurement results, peak value was obtained in a front position and the lowest at the 180° angle. The following is a concise and precise description of the results of the experiment. The complete and detailed information is available in [13].

### 3.1.2. EMF Exposure near the AS-RFID Reader

The distance was measured from the rear panel of the cabinet, where the RF emitting antennas are located, to the center of the antenna. The measurements at 135° and 180° have been made at angles of 45° and 90° angles, for the rear wall of the cabinet. The heights were measured from the floor to the center of the antenna. Figure 5 presents the map where the measurement points are marked.

At 1.0 m, 1.3 m and 1.7 m heights, the average of the recorded E-field strength values was 0.056 V/m, 0.059 V/m and 0.061 V/m, respectively.

Along with the distance increase, there the E-Field decrease was recorded, that was more specific and more clearly observable as the measuring angle was getting narrower, the foregoing can be up to almost 10 dB/m at 4.0 m for zero angles. However, at 180° we can observe more constant values.

The test was also carried out for the same 866 MHz frequency value, inside the cabinet at different heights of from 0.61 and 1.65 m, above the cabinet floor. Measures were also carried out at 0.075 m from the rear panel, where the emitting equipment was located, and at 0° angle. The field levels remained at approximately 0.270 V/m for all heights applied for the purpose of the test. The results obtained are presented as the Radiation Diagram in Figure 7 and the results presented on a 2D contour maps are accurately depicted in Section 3.5. later on.

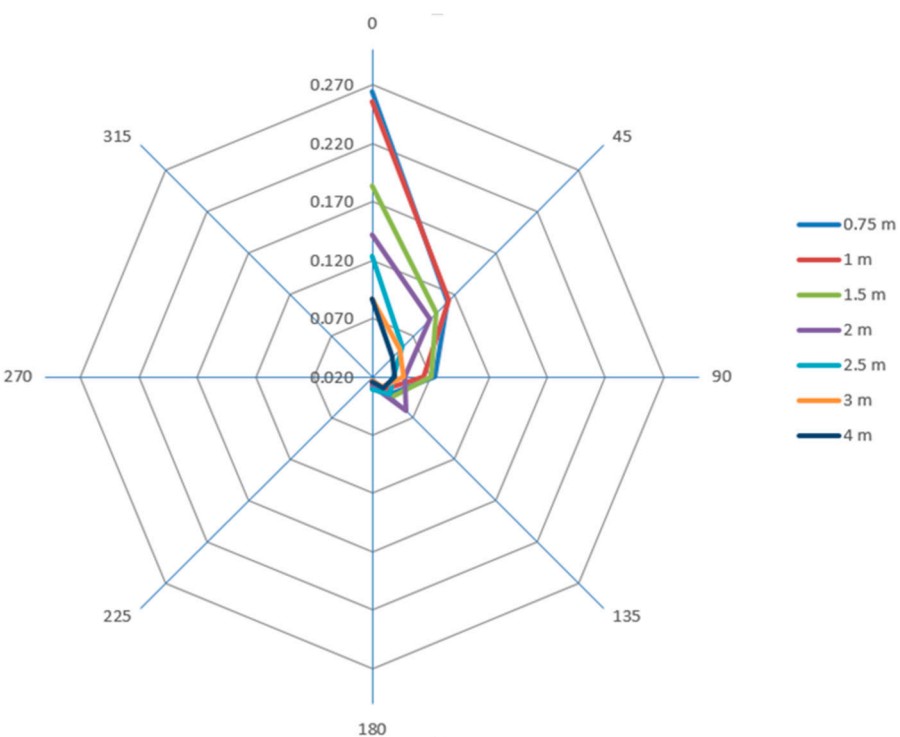

**Figure 7.** Spatial distribution of E-field measured near AS-RFID reader (V/m).

### 3.2. Anechoic Chamber—Madrid (Evaluation by Measurements)

#### 3.2.1. EMF Exposure near MO-RFID Reader

The dimensions of an anechoic chamber, where measurements on the EMF emitted by the MO-RFID reader were made, were as follows: 9.76 m × 6.71 m × 6.10 m. The reader was set on a manual positioning device, and, thus, the former could be rotated to facilitate the measurement of the radiation pattern. The radiation pattern, (maximum values from measurements for horizontal and vertical positions of measuring antenna and adjusted RFID antenna position) was sampled out around antenna, starting in front of an UHF RFID antenna with a 45° angle step, to cover major, lateral and, back radiation.

The differences in E-field values measured (Em) and calculated (Ec) are shown in Table 6 and Figure 8. The calculated measurements are obtained based on the ERP (Effective Radiated Power) values, and an equivalent calculation of the E-field intensity is performed at the measurement distance. The test equipment, with a transmission button pressed, delivers emissions for an approximate time of 1 min and then emission stops.

**Table 6.** The results of measurements and calculations of E-field near MO-RFID reader (at 3 m away).

| Azimuth | Ec | Ec ± U | Em | Em ± U |
|---|---|---|---|---|
| (o) | V/m | V/m | V/m | V/m |
| 0 | 1.30 | 0.89–1.89 | 1.96 | 1.26–3.04 |
| 45 | 1.30 | 0.89–1.89 | 1.79 | 1.16–2.78 |
| 90 | 1.08 | 0.74–1.57 | 1.39 | 0.89–2.15 |
| 135 | 0.95 | 0.65–1.38 | 0.95 | 0.61–1.48 |
| 182 | 0.34 | 0.24–0.50 | 0.46 | 0.30–0.72 |
| 225 | 0.33 | 0.23–0.49 | 1.08 | 0.70–1.67 |
| 270 | 0.53 | 0.37–0.77 | 1.44 | 0.93–2.24 |
| 315 | 1.04 | 0.71–0.71 | 1.79 | 1.16–2.78 |

Ec—E-field calculated value; Em—E-field measured value; Ec ± U—the range of E-field calculated ± uncertainty; Em ± U—the range of E-field measured ± uncertainty.

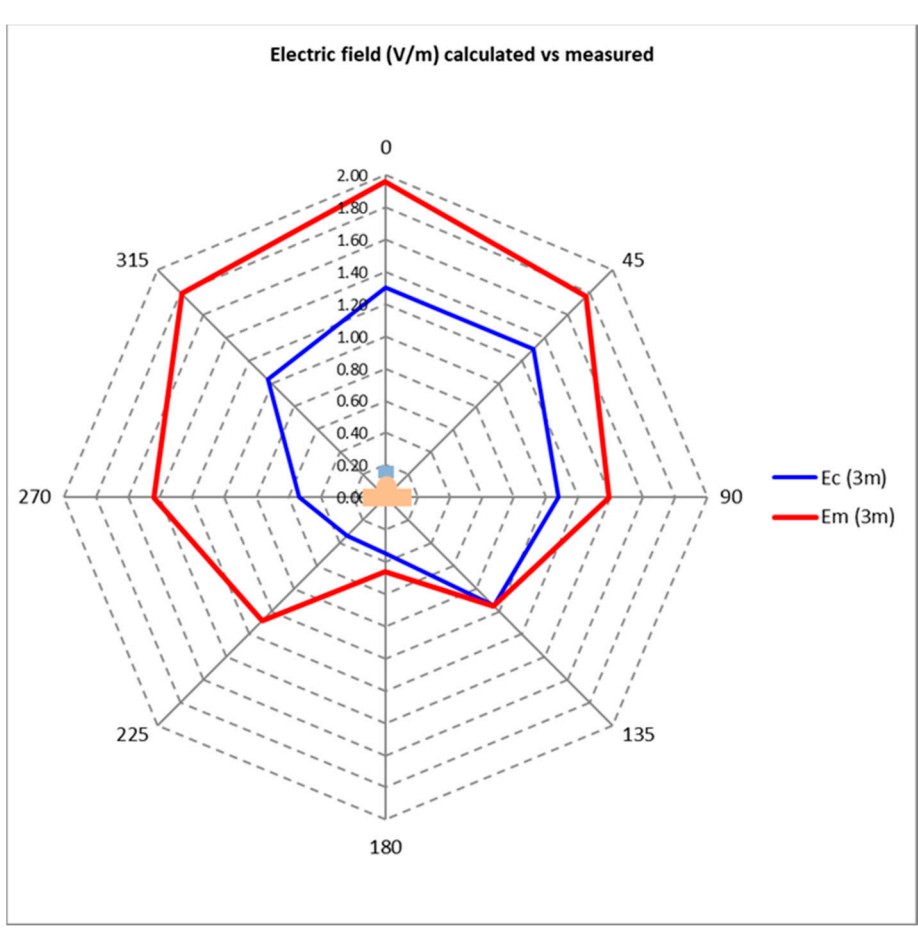

**Figure 8.** Spatial distribution of E-field measured and calculated near MO-RFID reader.

### 3.2.2. EMF Exposure near AG-RFID Reader

The maximum levels, 15.88 V/m and 10.68 V/m, were registered at the shortest as can be seen. These values decrease notably along with the distance. The lowest levels were registered at 3 m away from AG-RFID reader, with values of 1.49 V/m and 0.71 V/m recorded at 0° and 45° angles, respectively, and 0.25 V/m at of from 90°–270°. Table 7 presents the E-field strength values along the direction of the maximum radiation from the device under test in function of the distance. Figure 9 presents the calculated radiation pattern of the AG-RFID reader, depending upon the distance value [13].

**Table 7.** The E-field peak values measured at 0.5 m, 1.0 m and 3.0 m from the AG-RFID reader [13].

| | Calculated E-Field, Ec, V/m | | |
|---|---|---|---|
| Azimuth, (o) | 3 m Away from the Reader | 1 m Away from the Reader | 0.5 m Away from the Reader |
| 0 | 2.65 | 7.94 | 15.88 |
| 45 | 1.78 | 5.34 | 10.68 |
| 90 | 0.51 | 1.52 | 3.05 |
| 135 | 0.44 | 1.33 | 2.65 |
| 182 | 0.39 | 1.17 | 2.33 |
| 225 | 0.32 | 0.96 | 1.91 |
| 270 | 0.69 | 2.06 | 4.11 |
| 315 | 20.3 | 6.08 | 12.16 |

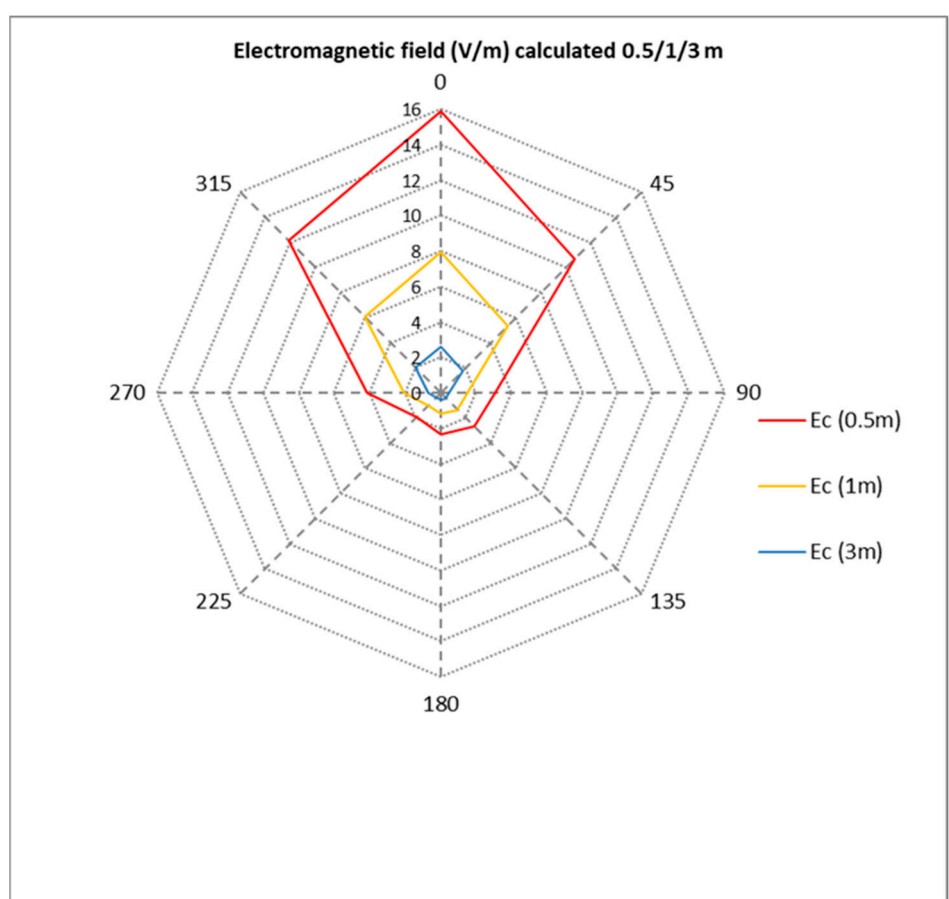

**Figure 9.** Spatial distribution of E-field calculated near AG-RFID reader [13].

A concise and precise description of the experimental work is presented here. Additional information is available in [13].

### 3.3. Specific Absorption Rate (SAR) Values from Exposure near Handheld UHF-RFID Readers

Considering the exposure to EMF of the user of the MO-RFID reader, the highest WB SAR value (0.036 W/kg, i.e., approximately to 50% of relevant limit regarding the exposure of general public, as set by the European recommendation 519/1999) was obtained at the AU and BU exposure scenarios, when a MO-RFID reader is grasped near the front of the body. The obtained SAR values were summarized in Table 8.

Considering the exposure of the other person present nearby the activated reader (scanned person or anyone present in the vicinity), the highest WB SAR value (0.034 W/kg,

again approximately to 50% of the limit on public exposure) was obtained at the CP exposure scenario, when MO-RFID reader is in front of the chest.

**Table 8.** Whole body averaged and local SAR under exposure to an EMF at a frequency of 865 MHz emitted by the UHF-RFID gun reader; values normalized at level of the EMF emission from an RFID at 1 W (ERP), (6 min continuous exposure).

| Exposure Scenario | WB NSAR | NSAR 10g | | | | |
|---|---|---|---|---|---|---|
| | | Head | Torso | Palm | Arm | Leg |
| AU–Head-20 cm | 0.036 | 0.091 | 0.038 | 4.2 | - | 0.0026 |
| BU–Chest-20 cm | 0.036 | 0.054 | 0.34 | 3.8 | - | 0.0079 |
| CP–Hip-16 cm | 0.034 | 0.33 | 2.1 | - | 0.15 | 0.051 |
| DP–Chest-5 cm | 0.014 | 0.095 | 0.82 | - | 2.1 | 0.020 |
| EP–Side-5 cm | 0.022 | 1.3 | 0.46 | - | 0.15 | 0.0021 |
| FP-Chest-5 cm | 0.018 | 2.1 | 0.15 | - | 0.014 | 0.0044 |

The maximum values of local SAR 10g were obtained for the palm of MO-RFID reader operator (in the AU scenario), as well as the breast and head of the exposed person (in the CP and FP scenarios)—at the levels comparable with a relevant limit regarding the exposure of general public. Local SAR 10g values in other analyzed cases (in legs, torso and head) were found significantly lower due to significantly greater distance between the reader antenna and the particular parts of the body.

Figure 10 shows the E-field distribution within the UHF-RFID gun area, at a radiated power of 1 W: distribution along the plane normal to the antenna.

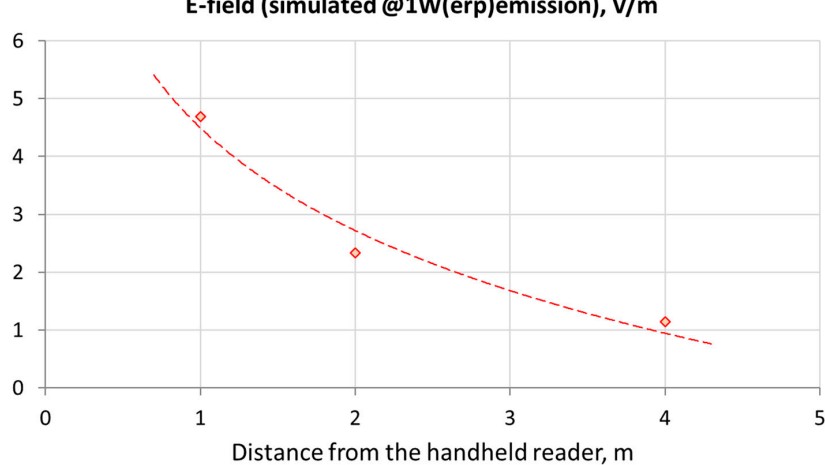

**Figure 10.** E-field distribution at a plane normal to the UHF-RFID gun antenna plane at a radiated power of 1 W. Spatial distribution of E-field measured near the AS-RFID reader.

The reading range of various UHF-RFID reader depends upon the radiated power and sensitivities of a passive tag used (typically 0.6–1 V/m of EMF exposure coming from the reader is required to read passive tags, however older tags may have lower sensitivities) [12]. The passive tags reading range of such sensitivities at a radiated power of 1 W was found in the range 4–10 m. The E-field values lower than 3 V/m (minimum required electromagnetic immunity of medical equipment used in professional healthcare environment according EN 60601-1-2) in this case were found at distances longer than approximately 30% of the reading range [35].

The worst-case may be depicted as a stay near a reader for 6 min with continuous emission at maximum power, although a more usual exposure scenario is with a shorter exposure time. Moreover, the remaining furniture, its fixation, the gun handle and the

monitor were not considered in our model, but they may also contribute to the decrease of the EMF values. However, RFID guns may also have emitting antennas for other radio communication technologies operating at radiofrequency band, like Bluetooth, Wi-Fi, public cellular networks (used for the transmission of data, e.g., using LTE services), which can increase the SAR in a human body when activated along with the gun RFID antenna. The following is a concise and precise description of the results of the experiment. Additional information is available in [12].

### 3.4. Simulations of E-Field Distribution from AG-RFID Reader Using In-House 3D-RL Algorithm

E-field distribution within cut plane heights is presented in Figure 11. The maximum values are obtained inside the room in which the AG-RFID reader is located. It was positioned on the center of a room ceiling, with a height around 3.5 m, covering an estimated zone of 8 × 8 m. As height increases, so do the average E-field levels, as the measuring points are closer to the reader antenna.

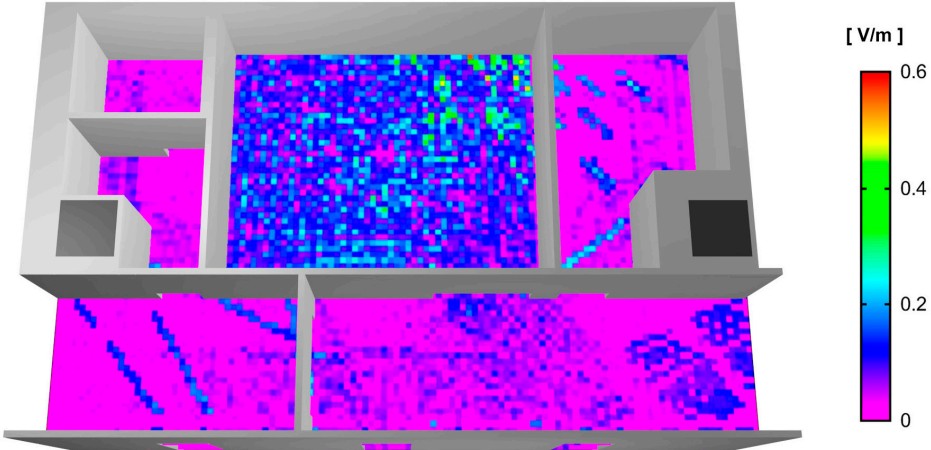

**Figure 11.** E-field distribution within the hospital ward simulation scenario [13].

To validate the E-field results obtained with the 3D RL approach, they have been compared with those from the ward scenario in the real case, considering different angle values (with orientations of 0°, 45°, 90° and 135°) and different cut-plane heights (h = 1 m and h = 1.7 m) inside the room. Some results are displayed in Figures 12 and 13, showing good agreement between the simulation and measured levels, within a linear radial distribution, 0° orientation, and cut plane height of 1 m. Deviations between the simulation and measurement results were in the range of 2–5% in E-field value estimation, and are given by differences between the topo-morphological description of the simulation scenario as compared to the real one, as well as deviations in material parameters.

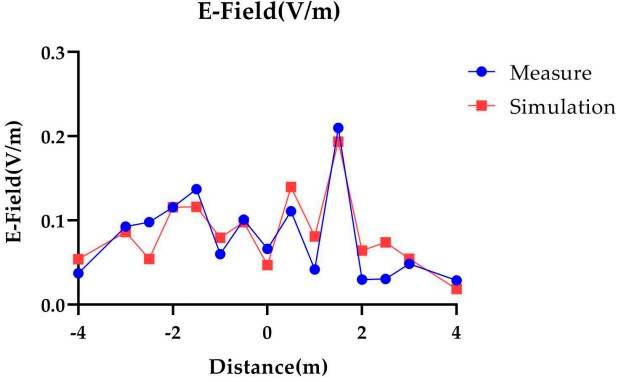

**Figure 12.** E-field estimation within a linear radial distribution, 0° orientation, cut plane height 1 m [13].

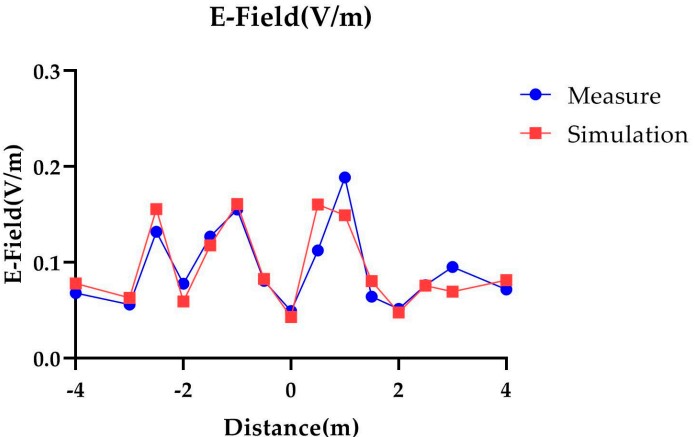

**Figure 13.** E-field estimation within a linear radial distribution, 0° orientation, cut plane height 1.7 m [13].

A concise and precise description of the experimental work is presented here. Wide and detailed information may be found in [13].

### 3.5. 2D Contour Maps Representation of E-Field Distribution near AS-RFID Reader

The graphic designs are to be perceived to be an accurate view of the EMF values at specific locations at the healthcare center. After the data collecting process is finished, the data were transferred to the graphic software (Surfer 8) to generate 2D contour maps, based on the previously measured EMF parameters. For each measurement, the antenna was oriented to detect the E-field maximum level and make relevant records. The EMF value curves were accurately presented. Graphs were prepared to depict the measurements at three different heights in the case of AS-RFID reader. As in the case of the AG-RFID reader, the curves of the E-field distribution were included there, as well. Given the fact that this distribution is symmetric, the results were extrapolated to the other unmeasured side. The 2D contour maps and the location of the radiation source are shown in Figure 14a–c. The mean values obtained are 0.056 V/m for 1.0 m high, 0.059 V/m for 1.3 m high, and 0.061 V/m for 1.7 m high. The points with the higher electromagnetic exposure recorded are accordingly marked in a more intensive red color, specifically in the front part, which is the direction where the radiating elements of the rear part are pointing.

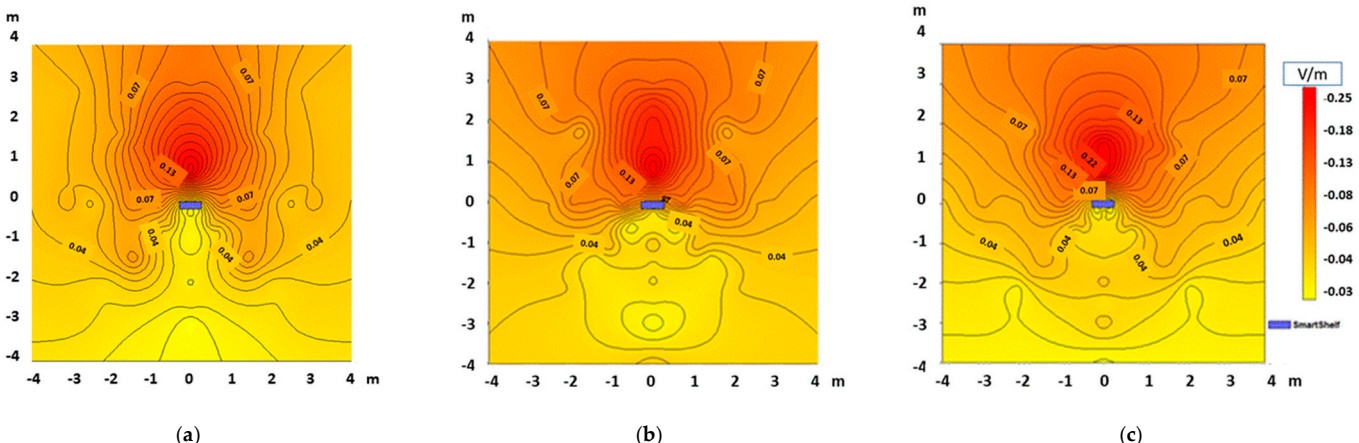

**Figure 14.** 2D Contour map of the experimental values of the E-field (V/m) in the vicinity, up to 4.0 m from the center point, of AS-RFID reader: (**a**) at 1.0 m height above the floor; (**b**) 1.3 m height above the floor; (**c**) 1.7 m height above the floor.

## 4. Discussion

Given the more and more frequent use of the wireless networks within the proximity of medical devices, it will be crucial to determine the points with higher exposure levels. Following the results obtained by the authors, the EMF measured in far-field (sufficiently away from the readers, i.e., at distance of at least 0.5 m away from the readers antenna) are within the limits provided to evaluate the health hazards to exposed humans, that have been due to the thermal effects of the absorption of electromagnetic energy in tissues, as specified for in the applicable laws and regulations and research review reports. At the frequency of 866 MHz, like for the EMF emission from the tested UHF RFID readers, the specific limits aimed to assess the thermal exposure effects, lie within the range of 40–200 V/m, for example: the limit of 88.3 V/m has been delivered to assess the EMF exposure at the work environment, in compliance with the European Directive 2013/35/EU, [36] or the limits of 88.3 V/m and 193.8 V/m were set forth in the ICNIRP 2020 to assess whole body and local occupational exposure, respectively, or accordingly, many-fold lower limits set for the general public exposure at discussed frequency.

Nevertheless, what should be emphasized to this extent is the fact that the level of emitted EMF is inversely proportional to the distance from the emitting antenna, and to the reading range of the applied RFID system. Consequently, the closer to the antenna, the more the EMF level rises. At the near-field, within a distance shorter than the wave length (as for the consider RFID readers emitting EMF of an approximate wavelength of 0.3 m), the thermal effects of EMF exposure near the source are to be examined, with the use of SAR computer simulations. The results of the studies carried out for such near-field exposure case (when a manually operated RFID reader is used, or by mistake rather, anyone is present directly close the autonomous reader) show that the electric field strength directly close to the reader may raise significantly exceeding 5 V/m, when in the far-field the EMF exposure level is many-time lower, depending on the distance from the antenna and the reading range of used system. With regard to the average value of the measured E-field, the distances at which these values were obtained and the fact that the exposure would be sporadic near autonomous readers located away from people, the conclusion goes that an operator of the considered manually operated devices or the individuals approaching their proximity may suffer from the exposure at levels comparable to the exposure limits provided to protect against thermal effects of EMF influence on exposed tissues. The results obtained by the authors show that whilst taking the EMF exposure effects in the near-field surrounding UHF RFID readers into consideration, the relevant preventive measures should be undertaken with the aim being to ensure sufficient protection of individuals who may be present there, where the RFID reader emitting over 1 W of electromagnetic radiation is used and the individuals may access the areas that is closer than 30% of reading range of the applied RFID system.

However, all guidelines included the notes that the vulnerable individuals may require additional protection, including also the protection against inappropriate operation of electronic devices, at the level of exposure which is formally compliant with international exposure limits—with the foregoing to be of specific significance for medical implants that are more often used in healthcare centers than in public areas. Specifically more attention should be paid to the use of manually operated RFID the application of which is planned at the areas accessible to patients and visitors. The foregoing implies the E-field levels may be considered negligible where autonomous readers are used and located far from space accessible for workers, patients and visitors in healthcare centers. When individuals are within a direct proximity of readers emitting antennas, there emerges a need to undertake the relevant prevention measures, that will be of assistance to vulnerable individuals, users of medical implants and electronic devices. To this extent, the lowest limit of E-field to be examined as regards the electromagnetic interferences with electronic devices (EMC) is 3 V/m (CENELEC) EN 60601-1-2:2015 [35]. Therefore, the space where E-field level may exceed the EMC-related attention level is many-time larger than the space of exposure which requires evaluation with respect to SAR limits. However, following the SAR results,

it is revealed that the EMF exposure near to the manually operated UHF RFID readers (guns) do not provide the SAR values that will exceed the general public limits, on the condition that the emission does not exceed 1 W, which in fact implies the use of the RFID system with a reading range of from 3 to 10 m (depending upon the sensitivities of used tag).

The E-field values obtained with the use of the 3D-RL deterministic approach showed good agreement between the simulation and measurement values, indicating the adequate option to perform estimation of volumetric E-field distribution and mapping with the use of the proposed methodology. This is why the foregoing approach is considered a useful tool, with the specified methodology to provide an immediate and accurate vision of the EM fields, for detect the highly exposed areas, within the proximity of the patients, visitors and the healthcare personnel. The 2D graphic representation, with 2D contour maps, may be of assistance when ensuring protection against EM interferences with medical devices, that may cause potential damage to patients, and when detecting highly exposed locations, whereby the emission values do not reach the suggested limits, and when planning and developing the prospected high-tech healthcare centers [6]. All the above specified activities and measures should be compatible with a sufficient signal level set for the wireless systems as installed at a healthcare center.

A potentially excessive amount of exposure to the EMF emitted by UHF-RFID devices can be dangerous to patients or other people; therefore, it requires further more comprehensive analysis and examination, with the good practice guidelines to be accurately developed for the design and use of RFID systems at the healthcare industry.

There have been some limitations for the exposure formally compliant with mentioned international guidelines, including the principles to be applicable to precautionary measures that in fact require some further consideration [37,38]. The limits provided by the guidelines are considered only with regard to the protection against the thermal effects of such exposure, yet not against any other potentially harmful effects as reported for the lower EMF exposures. However, as it has been determined under the applicable guidelines and laws and regulations, vulnerable individuals, such as the users of medical implants specifically, may require some further protection to be further considered, for example as required from employers' activities, following the provisions of the Directive 2013/35/EU [36].

The electromagnetic hazard assessment methods subject to discussion hereunder, as well as the issue of the assessment of the RFID system operation in healthcare environments, seems to be of importance due to social issues, legal issues as well as scientific research that are continuously applicable to such assessment, e.g., following the rhetoric of the SCHEER Scientific Committee on Health, Environmental and Emerging Risks, elaborating their opinion on the need of a revision of the annexes in Council Recommendation 1999/519/EC and Directive 2013/35/EU, in view of the latest scientific evidence available with regard to RF (100 kHz—300 GHz)—provided for public discussion in 2022.

## 5. Conclusions

Due to the more frequent use of the wireless communication networks within the proximity of medical devices at healthcare centers, it has become crucial to determine the points with higher levels of exposure.

A comprehensive exposure assessment, source identification, and configuration at maximum transmitted power or actual maximum power is performed. The characteristic of Radiofrequency EMF exposures have been studied, in terms of their spatial distribution denoted at the E-field strength, based upon the measurements or calculations results.

Following the results obtained by the authors, the electric field strength (at 866 MHz frequency) values, as measured or simulated with the use of the computer modelling in far-field areas of the examined UHF RFID readers intended for manual or autonomous operation at the healthcare centers (>0.5 m from antenna), are substantially lower than the exposure limits provided for in guidelines, laws and regulations that are applicable to the

protection of health of the EMF exposed individuals against hazards associated with the thermal effects of the absorption of electromagnetic energy in tissues.

Still, all the guidelines included notes whereby it was stated that vulnerable individuals may require some additional protection at exposure levels below mentioned thermal effects-based limits, including the protection against inappropriate operation of electronic devices, with the foregoing of specific importance for the medical implants that appear to be more likely used in the healthcare centers than in public areas.

The evaluation of the thermal effects of EMF exposure near the readers (in case of exposure in near-field surrounding of EMF source—when manually operated UHF RFID reader is used, or when accidental exposure of an individual who is approaching an autonomous reader) requires the SAR computer simulation. The results obtained by the authors shown that whilst taking the effects of the EMF exposure in the near-field surrounding UHF RFID readers into account, the relevant preventive measures are to be undertaken, with the aim being to ensure sufficient prevention of individuals who may be present there. Applying relevant prevention measures needs to be considered specifically where the RFID reader that emits electromagnetic radiation of over 1 W has been used, or where the vicinity of applied RFID system is accessible for individuals, who may approach to the reader closer than 30% of its reading range.

To conclude, the use of manually operated RFID readers should specifically constitute the focal point of further research on the matter, as well as their prospected installation at the areas accessible by patients and visitors.

**Author Contributions:** Conceptualization, V.R., O.J.S. and J.A.H.; methodology, V.R., O.J.S., J.K., F.F. and J.A.H.; software, S.S., V.M.F., E.A., M.CE., L.E.R., P.M. and P.Z.; validation, V.R., O.J.S., J.A.H., J.K. and F.F.; formal analysis, O.J.S., E.A., P.Z., M.C.-E. and J.A.H.; investigation, V.R., J.A.H., F.F. and J.K.; resources, V.R., O.J.S. and J.A.H.; data curation, O.J.S. and P.M.; writing—original draft preparation, V.R., J.A.H., O.J.S., S.S., E.A., P.Z., L.E.R., M.C.-E. and P.M.; writing—review and editing, V.R., O.J.S. and J.A.H.; visualization, V.R.; supervision, V.R.; project administration, V.R.; funding acquisition, V.R. All authors have read and agreed to the published version of the manuscript.

**Funding:** This work was supported by Instituto de Salud Carlos III project "Electromagnetic "Characterization in Smart Environments of Healthcare, and their involvement in Personal, Occupational, and Environmental Health" (PI14CIII/00056) https://portalfis.isciii.es/es/Paginas/DetalleProyecto.aspx?idProyecto=PI14CIII%2f00056 (accessed on 24 July 2022), and project " (PI19CIII/00033) TMPY 508/19 " Metrics development for electromagnetic safety assessment in healthcare centers in the context of 5G" https://portalfis.isciii.es/es/Paginas/DetalleProyecto.aspx?idProyecto=PI19CIII%2f00033, (accessed on 24 July 2022) from Sub-Directorate-General for Research Assessment and Promotion. The results of a research task (II.PB.15) carried out within the National Programme "Improvement of safety and working conditions" partly supported in Poland in 2020–2022—within the scope of research and development—by the National Centre for Research and Development were also included.

**Informed Consent Statement:** Not applicable.

**Data Availability Statement:** Data supporting reported results can be found at https://repisalud.isciii.es/handle/20.500.12105/8365 (accessed on 24 July 2022).

**Acknowledgments:** Authors special acknowledge for his guidance and all contributions of José Carlos Fernandez de Aldecoa at Hospital Universitario de Canarias, to David Rubio from the Secretary of State for Digital Progress, General Directorate of Telecommunications and Technologies of the information (Madrid, Spain), for his contribution in performing measurements in the semi-anechoic chamber, and to Irene Perán for her technical support with AUTO-CAD works.

**Conflicts of Interest:** The authors declare no conflict of interest.

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
