# Peer review of "Electromagnetic Assessment of UHF-RFID Devices in Healthcare Environment"

_applsci, doi:10.3390/app122010667_

Round 1

Reviewer 2 Report

This a competent piece of work, but let down a little by the presentation

The detail on measurements and SAR calculation is sparse and although I am confident that they are correct, the methodologies do need more detail.

The reason why exposures are not compared with the Workers' EMF Directive  need to be more fully explained. There is a short comment about workers being uninformed, or not directly employed as EMF workers, but this means that the lower tier of the EMF Directive applies, not ICNIRP. 

Similarly, and device which can produce public exposures already should have been assessed before they were  put on the market. 

Author Response

Respond to Reviewer 2 Comments

This a competent piece of work, but let down a little by the presentation.

- The detail on measurements and SAR calculation is sparse and although I am confident that they are correct, the methodologies do need more detail.

                Response: We thank the reviewer comment. In the revised manuscript we’ve provided more detailed information regarding the performed measurement and SAR calculations – mostly in the Method section.

Motivation for substantial redrafting of the manuscript is provided in the response to the Reviewer 1.

- The reason why exposures are not compared with the Workers' EMF Directive  need to be more fully explained. There is a short comment about workers being uninformed, or not directly employed as EMF workers, but this means that the lower tier of the EMF Directive applies, not ICNIRP. 

                Response: We thank the reviewer comment. The application of particular exposure limits depends on the context of considered exposure scenarion – it was explained in the revised manuscript in additional sections included in the Introduction section, as well as mentioned in the Discussion section in the revised manuscript.

- Similarly, and device which can produce public exposures already should have been assessed before they were  put on the market. 

Response: We thank the reviewer comment. Similarly, to the limits regarding humans exposure, also the application of particular limits regarding the technical requirements on EMF emission and immunity depends on the context of considered exposure scenarion/the environment of the intended use of devices  – it was explained in the revised manuscript in additional sections included in the Introduction section, as well as mentioned in the Discussion section in the revised manuscript. We were not discussed "CE" mark-related requirements (applicable in general for the general public environment – when the healthcare environment is managed by different regulations) in details – but we do hope that improvements in the revised manuscript explain sufficiently issues rised by the Reviewer.

Round 2

Author Response

 We thank the reviewer comment, which provide an improvement to the document.

Section 2.4. Text is revised and improved by means of reduced duplicate rate. Some information that is not relevant to the contents of the manuscript is deleted. It is included Figure 6, with included AU, BU, CP, DP, EP and FP.

Section 3, Results. Revised presentation of results. – All Section 3 has been depthly reviewed, by a native speaker.

Section 3.1.2.  Text revised and some information reduced, specially which is not relevant to the contents of the manuscript. Figure 7 is modificate taking into account text Units: (V/m) instead of (dB) and (m) instead of (cm), in order to unify all units in the revised manuscript.

Section 3.3. Text revised, some information not relevant to the contents of the manuscript reduced and duplicate rate also reduced, explicitly indicated in the revised manuscript version.
